# Loss and Gain of Gut Bacterial Phylotype Symbionts in Afrotropical Stingless Bee Species (Apidae: Meliponinae)

**DOI:** 10.3390/microorganisms9122420

**Published:** 2021-11-24

**Authors:** Yosef Hamba Tola, Jacqueline Wahura Waweru, Nelly N. Ndungu, Kiatoko Nkoba, Bernard Slippers, Juan C. Paredes

**Affiliations:** 1International Centre of Insect Physiology and Ecology, Nairobi 30772-00100, Kenya; yosef@icipe.org (Y.H.T.); jackiemaingih@gmail.com (J.W.W.); nndungu@icipe.org (N.N.N.); nkiatoko@icipe.org (K.N.); 2Department of Biochemistry, Genetics and Microbiology, Forestry and Agricultural Biotechnology Institute (FABI), University of Pretoria, Pretoria 0002, South Africa; bernard.slippers@up.ac.za

**Keywords:** *Lactobacillaceae*, *Bifidobacteriaceae*, *Acetobacteraceae*, *Meliponula*, *Dactylurina*, *Hypotrigona*, *Liotrigona*

## Abstract

Stingless bees (Apidae: Meliponini) are the most diverse group of corbiculate bees and are important managed and wild pollinators distributed in the tropical and subtropical regions of the globe. However, little is known about their associated beneficial microbes that play major roles in host nutrition, detoxification, growth, activation of immune responses, and protection against pathogens in their sister groups, honeybees and bumble bees. Here, we provide an initial characterization of the gut bacterial microbiota of eight stingless bee species from sub-Saharan Africa using 16S rRNA amplicon sequencing. Our findings revealed that Firmicutes, Actinobacteria, and Proteobacteria were the dominant and conserved phyla across the eight stingless bee species. Additionally, we found significant geographical and host intra-species-specific bacterial diversity. Notably, African strains showed significant phylogenetic clustering when compared with strains from other continents, and each stingless bee species has its own microbial composition with its own dominant bacterial genus. Our results suggest host selective mechanisms maintain distinct gut communities among sympatric species and thus constitute an important resource for future studies on bee health management and host-microbe co-evolution and adaptation.

## 1. Introduction

More than 90% of tropical plant flowers are pollinated by animals; among them, bees are the most important pollinators [1,2,3]. Stingless bees (Meliponinae) represent the most diverse group among corbiculate bees and are commonly distributed in the tropical and subtropical regions; 77% are found in the Neotropics (Central America, Caribbean, and South America), followed by 16% in the Indo Malay/Australasian region and 7% in the Afro tropics [4,5,6,7,8,9]. Worldwide, there are approximately 61 identified stingless bee genera and over 550 species [4,8,10]. In Africa, there are 10 genera and 36 species of stingless bees described, and they are commonly distributed in tropical forests, savannahs, and deserts [4,10,11]. In Kenya, 6 genera (*Cleptotrigona*, *Dactylurina*, *Hypotrigona*, *Liotrigona*, *Meliponula*, and *Plebeina*) have been reported and are composed of 12 species that inhabit the Kakamega forest, Mwingi, the Arabuko-Sokoke Forest, and the Taita hills [10,12,13,14]. The *Meliponula* genus has been further divided into three sub-genera, namely *Meliponula* (one species), *Axestotrigona* (two species), and *Meliplebeia* (five species) [6,10].

The current increasing demand for managed pollinators and the advantages that stingless bees present have further encouraged their domestication [15,16,17,18,19]. In contrast to honeybees, stingless bees do not sting, they can be used for crop pollination in greenhouses, and they produce honey, propolis, and wax with important medicinal properties (e.g., antioxidant capacity, antimicrobial activity) of high economical value [17,20,21,22,23].

Bacterial symbiosis is a profound mechanism that has shaped insect physiology and ecology throughout their evolution [24,25]. With the increased number of insect models used for beneficial gut bacterial studies, there is cumulative evidence that insect gut microbiota play an important role in insect physiology, organ development, protection against natural enemies and help in their nutrition and detoxification. Additionally, they are involved in immune response activation and communication [26,27,28,29,30,31]. Bee gut bacterial communities are not an exception [32,33,34,35,36,37,38,39].

The vast majority of bee gut bacterial microbiota research has been conducted on honeybees (Apini tribe), and to a lesser extent, in bumble bees (Bombini tribe), but very little is known regarding stingless bees [40,41,42,43,44,45]. The handful of studies investigating stingless bee gut bacterial microbiota have reported substantial community diversity that varies across bee species [40,42,43,44,45,46]. *Lactobacillus* and *Bifidobacterium* have been reported as the most abundant genera, and in contrast to Apini and Bombini tribes, in Meliponini, *Gilliamella* and *Snodgrassella* are not always present or abundant. Additionally, stingless bees harbor numerous and specific *Acetobacter* species [41,42,43,44,45,46,47].

Despite being the most diverse group of corbiculate bees, only a few stingless bee species have been studied for their gut bacterial microbiota. Thus, much greater diversity is expected since they occupy a variety of tropical and subtropical habitats, and they present a large range of behaviors, such as, specific plant foraging and the construction of perennial nests that have different locations and structures [4,10,11,18,48]. Therefore, a better characterization of stingless bee gut bacterial microbiota will contribute to understanding host–symbiont interactions which likely impact stingless bee physiology. This will lay the foundation for different strategies to ensure beneficial microbe persistence in bees for agricultural sustainability and biodiversity conservation.

Here, we characterized the gut bacterial microbiota of eight different Afrotropical stingless bee species belonging to *Meliponula*, *Dactylurina*, *Hypotrigona*, and *Liotrigona* genera and evaluated their phylogenetic diversity and their relationship with other corbiculate bees from other continents.

## 2. Materials and Methods

### 2.1. Sample Collection and Preparation

Foragers from eight African stingless bee species belonging to four genera (*Meliponula*, *Dactylurina*, *Hypotrigona*, *Liotrigona*) were collected from colonies located at the International Centre of Insect Physiology and Ecology (*icipe*) meliponary, Duduville campus, Nairobi, Kenya, from October 2019 to February 2020. The following bees were originally collected from the Kakamega forest (Western Kenya): *Meliponula bocandei* (collected in February 2019), *Meliponula togoensis* and *Meliponula ferruginea* (collected in September 2018), and *Meliponula lendliana* and *Liotrigona* spp. (collected in September 2019). From the Mwingi (Eastern Kenya), the following bees were collected: *Hypotrigona* spp. 1 (collected in February 2017) and *Hypotrigona* spp. 2 (collected in August 2015). From the Taita Hills and Kilifi (Coastal Kenya), *Dactylurina schmidti* were collected (collected in September 2018). At the *icipe* meliponary, all colonies had access to the same floral resources and faced the same ecological conditions (e.g., temperature, humidity). Gardens in their foraging range included *Acacia*, *Eucalyptus*, *Psidium*, *Cordia Africana*, *croton megalocarpus*, *Leucaena glauca*, and mixed crops.

Fifty forager worker bees per hive from two hives per species were collected at the hive entrance and washed in 4% sodium hypochlorite, followed by a wash in 70% ethanol and finally 1× PBS for 2 min to eliminate any external microorganisms or contaminant DNA attached to the cuticle [49,50]. The entire gut, with its contents, was dissected aseptically from each bee using forceps, and the guts were placed in a 2 mL microcentrifuge-tube containing 500 µL PBS. Samples were stored at −80 °C before DNA extraction.

### 2.2. DNA Extraction

DNA extraction from the entire gut was done using the CTAB-Phenol–Chloroform extraction method described in [50]. DNA was resuspended in 200 µL sterile water, and DNA concentration and quality per sample was confirmed using NanoDrop 2000 Spectrophotometer (Thermo Scientific, Wilmington, NC, USA).

### 2.3. 16S rRNA Gene Amplification, Sequencing, and Gut Community Analysis

All 16S rRNA gene amplification and sequencing was done at Macrogen Europe BV (Meibergdreef 31, 1105 AZ Amsterdam, The Netherlands, https://dna.macrogen-europe.com/eng/, accessed on 17 April 2020). We analyzed 8,563,992 paired-end sequences spanning the V3-V4 region of the 16S rRNA with an average of 155,709 reads per sample (10 pooled guts per sample, and the number of reads ranged from 106,894 to 198,324). We used the DADA2 pipeline embedded in R (version 4.0.2) to analyze our sequence reads [51]. Reads were checked for quality and primer sequences trimmed using Cutadapt (version 2.10). Taxonomic classification was performed against the SILVA138 database using a pre-trained Naive Bayes classifier [52]. Taxa and ASVs with cumulative abundance below five were discarded from analysis and from our negative control (a “blank” sample was included in the run, and we recovered only 2 reads), as well as those classified as unwanted sequences of animal and fungal origin (Chloroplast, Chloroflexi, Mitochondria, Archaea, and Eukaryota). Data were rarefied for the downstream alpha and beta diversity analysis (Appendix A). Alpha diversity was determined using Evenness, Faith’s phylogenetic diversity, and the Shannon index, and the statistical differences of gut bacterial microbiota diversity across the species were tested using Kruskal–Wallis H test. Average diversity values were plotted for each sample at each even sampling depth and samples were grouped based on sample metadata. To visualize the clustering of the microbial communities per species, principle coordinate analysis (PCoA) was done using Bray–Curtis distances and UniFrac distances (Weighted and Unweighted UniFrac). A Pearson correlation test was additionally performed to identify the impact of bee size on the bacterial richness and evenness.

### 2.4. Phylogenetic Tree Analysis

Phylogenetic trees were constructed for *Lactobacillaceae*, *Bifidobacteriaceae,* and *Acetobacteraceae* members using MEGA X [53]. Sequences were aligned using Unweighted Pair Group Method with Arithmetic Mean (UPGMA). Phylogenetic trees were computed with the Kimura 2-parameter model using the Maximum Likelihood method [54]. The model was selected based on the Bayesian Information Criterion (BIC) values. The bootstrap consensus tree was inferred from 1000 replicates, and less than 50% bootstrap replicates collapsed.

## 3. Results

### 3.1. Bacterial Communities Associated with Stingless Bee Guts

We sampled eight different stingless bee species in Kenya: *Meliponula bocandei*, *M. togoensis*, *M. ferruginea*, *M. lendliana*, *Dactylurina schmidti*, *Liotrigona* spp., and *Hypotrigona* sp. 1 and sp. 2. We obtained 2038 amplicon sequence variants (ASVs). We found 409 genera of which 13 genera represented more than 99% of total reads (Figure 1A and Appendix A).

We found that Firmicutes, Proteobacteria, and Actinobacteria were the dominant phyla across the eight stingless bee species (Appendix A). The most abundant was Firmicutes with about 60% of total reads encompassing *Lactobacillus*, *Bombilactobacillus*, *Acetilactobacillus*, and *Apilactobacillus*, among others. *Lactobacillus* accounted for approximately 25% of total reads (Figure 1 and Appendix A). The second highest phylum was Proteobacteria with approximately 18% of total reads including *Saccharibacter*, *Neokomagataea*, *Wolbachia*, *Bombella*, *Gluconacetobacter*, *Nguyenibacter*, *Acinetobacter*, *Zymobacter*, *Klebsiella*, and *Acetobacter*. *Saccharibacter* accounted for about 8% of total sample reads (Figure 1 and Appendix A). The actinobacteria phylum including *Bifidobacterium* accounted for approximately 10% of total reads (Figure 1 and Appendix A). Interestingly, whereas almost all stingless bee species were dominated by a member of the *Lactobacillaceae* family, the dominant genera varied across species. *Acetilactobacillus* and *Apilactobacillus* related genera dominated in *Hypotrigona* sp. 2, *M. lendliana*, and *M. ferruginea*, *Lactobacillus* dominated in *Hypotrigona* sp. 1 and *M. togoensis*, and *Bombilactobacillus* in *D. schmidti*. *Bifidobacterium* was present in all species except *Hypotrigona* sp. 2 and was the most abundant in *Hypotrigona* sp. 1. Among the *Acetobacteraceae* family, *Bombella* was the most abundant in *M. bocandei,* and *Saccharibacter* was the most abundant in *D. schmidti* (Figure 1 and Appendix A). Interestingly, we also uncovered the endosymbiont, *Wolbachia,* which was highly abundant in all *Liotrigona* sp. (59% of *Liotrigona* sp. total reads) and in one sample of *M. lendliana* (0.5% of *M. lendliana* total reads) (Appendix A).

### 3.2. Bacterial Communities Varied across Stingless Bee Species

To measure the bacterial community variation across species, we analyzed the overall diversity of the gut bacteria in the eight stingless bee species (Figure 2). As seen in Figure 1, alpha diversity varied significantly when richness (Shannon: *p* = 0.000038, *H* = 29.672, (Figure 2A)), evenness (Evenness: *p* = 0.00005, *H* = 29.12, (Figure 2B)), and phylogenetic distances (Faith’s phylogenetic diversity, *p* = 0.00018, *H* = 24.888, (Figure 2C)) were tested. These results are consistent with the PCoA using Bray–Curtis and from the weighted and unweighted UniFrac distances (Figure 3).

Even though *M. lendliana*, *Hypotrigona* sp. 2, and *Liotrigona* sp. clustered together in Bray–Curtis, they did not cluster in Unifrac distances. It is also noteworthy that these are the species where we obtained a lower number of samples (2 to 4 compared to 8 to 10 for others (Figure 3)).

### 3.3. Phylogenetic Diversity of Stingless Bee Gut Bacterial Microbiota

To evaluate the relationship between the bacteria generated in this study to other bacterial sequences isolated in other corbiculate bees, we constructed phylogenetic trees with the three most abundant families, *Lactobacillaceae*, *Bifidobacteriaceae*, and *Acetobacteraceae* (Figure 4).

ASVs related to *Lactobacillaceae* were found overall distributed with other reported sequences from corbiculate bees. The most abundant ASVs were closely related to strains from *Lactobacillus* (Firmicutes 5 group) and *Bombilactobacillus* (Firmicutes 4 group), known members of honeybee and bumble bee core gut bacterial microbiota. Interestingly, we also uncovered multiple ASVs related to environmental or hive-associated *Lactobacillaceae* such as *Acetilactobacillus* and *Apilactobacillus* with an important abundance and prevalence across samples. One ASVs was related to *Ligilactobacillus* but at a very low abundance (Figure 1 and Figure 4).

*Bifidobacteriaceae* associated ASVs were found to cluster in two subgroups (Figure 5). The most numerous and abundant were highly related to *Bifidobacterium commune*, *Bifidobacterium bohemicum*, and *Bifidobacterium bombi*, all of which are found in bumble bee guts. Two other ASV clustered with *Bifidobacterium coryneforme* strains found in *Apis mellifera* and *M. bocandei,* from previous studies. Interestingly, none of the represented ASVs in Figure 4 were related to *Bifidobacterium asteroids* found in honeybees or bumble bees, but were found in the stingless bee *M. bocandei* from Kenya [44].

The *Acetobacteraceae*-associated ASVs clustered with other reported strains and in specific groups (Figure 6). The most abundant cluster was found to be highly related to other *Acetobacter* species isolated in other stingless bees, *Melipona* sp. and *Meliponula* sp., from Panama and Gabon [41,63,64]. We found one group also related to *Bombella apis* and *Bombella intestini* reported in honey bees and bumble bees, and another group with very long branches related to *Neokomagataea thailandica* isolated from flowers [41,65]. Interestingly, we found that some ASVs form a group on their own, with no closest known related strain (ASV 34, 35, 39, 52, and 75, Figure 6). In contrast to *Apis mellifera* bacterial microbiota, we did not uncover any *Commensalibacter* related strains in our study [66,67].

## 4. Discussion

We reported the first characterization of the gut bacterial microbiota of eight endemic African stingless bee species (Apidae: Meliponini). We found that Firmicutes, Actinobacteria, and Proteobacteria were the dominant and conserved phyla across the stingless bee species. Interestingly, there were significant variations amongst the genera that dominated the gut of each stingless bee species. Furthermore, phylogenetic analysis showed substantial bacterial strain diversity, which is shared with other corbiculate bees, but also with several African specific bacterial clusters. As has been reported for South American stingless bee species, the East African stingless bee species we examined lost the honeybee and bumble bee specific symbionts *Snodgrassella*, *Gilliamella*, and *Frischella* [42,46,69,70]. Together, these results are an important contribution towards the study of the common origin of corbiculate bee gut bacterial microbiota and their co-evolution with their host.

Among the African stingless bee populations that were sampled, we identified *Lactobacillaceae* (including *Lactobacillus*, *Paucilactobacillus*, *Bombilactobacillus*, and *Lentilactobacillus*), *Bifidobacterium*, and *Acetobacteraceae* (including *Saccharibacter* and *Neokomagataea*) as the most abundant families, which account for more than 73% of the total gut bacterial microbiota. Interestingly, phylogenetic analyses revealed novel diversity of strains from these groups that has not been reported in other studies on stingless bees [41,42]. Additionally, despite sharing the same habitat for at least six months, each stingless bee species was dominated by a certain number of specific bacterial strains. Evaluating bacterial gut composition across longer time scales—months or years—from feral colonies or after a change of location would be important to understand the causes of this diversity [40]. These results demonstrate that there is a vast diversity of stingless bee gut bacterial microbiota members that are yet to be characterized and that conducting research on multiple strains, as done in this study, is needed to have a comprehensive and comparative picture of stingless and corbiculate bee gut bacterial microbiota and their co-evolutionary patterns.

We found that gut bacterial microbiota richness and evenness varied across the stingless bee species. To get a better understanding of the drivers of bacterial diversity, we investigated potential correlations between bacterial diversity and bee size; however, we did not find any correlation (Appendix A). A recent study reported that larger bees have higher bacterial diversity, and that this correlation is higher when colony size was considered [42]. Since African stingless bee research is in its early stages, little is known about bee colony and hive size. This enables us to better calculate a potential correlation between bee habitat size and the gut bacterial microbiota diversity in their guts.

Whereas library sequencing of the 16S rRNA V3-V4 region has proved to be informative and very useful for bee bacterial gut microbiota characterization, it has important limitations in resolution and absolute bacterial quantification. Complementary studies evaluating whole bacterial genome sequences (DNA metagenomics) and bacteria metabolically active pathways (RNA metagenomics) will provide better information about bacterial classification and metabolic capabilities [36]. Additionally, such approaches will help to discriminate between metabolically active resident bacteria and transient inactive food-associated microbes.

Interestingly, we found the endosymbiotic bacteria *Wolbachia* in *Liotrigona* sp. with 100% prevalence that accounted for 59% of total reads, and in *M. lendliana* with 33% prevalence. The two *Wolbachia* ASVs found in this study were closely related to the *Wolbachia* strain WCAAL1 (MT590312, 97% similarity) which is found in the *Aedes albopictus* mosquito. Studies have shown that *Wolbachia* protects its host against viruses [71,72]. Since bees harbor a substantial diversity of viruses, it might be relevant to investigate the potential *Wolbachia* antiviral protection phenotype in these two stingless bee species [73]. Additionally, *Wolbachia* has also been shown to manipulate host reproduction by inducing parthenogenesis [74]. Most of stingless bee workers contribute to the production of haploid eggs, which generate male offspring or trophic eggs that can be eaten by the queen as a food source [75,76]. It would be interesting to investigate if there is any involvement of *Wolbachia* in these processes in *Liotrigona* spp. or *M. lendliana*.

Tropical regions are profitable environments to conduct gut bacterial microbiota studies due to their number of sympatric honeybee and stingless bee species and their diversity, which enables research on the evolutionary history of their bacterial microbiota [4,10,11]. This study contributes to the understanding of general concepts in microbial ecology and evolution in these bee species that could only be found in tropical latitudes and microbe rich environments.

## Figures and Tables

**Figure 1 microorganisms-09-02420-f001:**
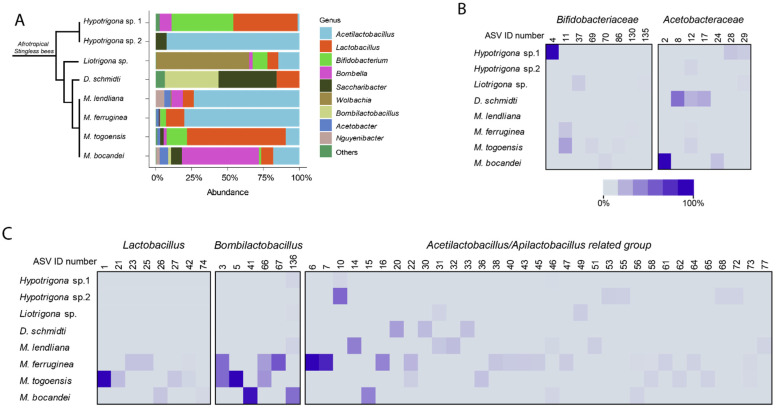
Gut bacterial genera associated with eight stingless bee species in Kenya. (**A**) A bar plot representation of all the stingless bee gut bacterial genera with an overall abundance higher than 1%. All genera with abundances below 1% were categorized as “Others”. The phylogeny of Afrotropical stingless bees was based on [8]. (**B**,**C**) heatmap comprising of the ASVs from the five most abundant genera across the eight stingless bee species. ASV ID numbers (Appendix A) are indicated at the top. Darker squares correspond to higher mean relative abundances for a given ASV in each bee species.

**Figure 2 microorganisms-09-02420-f002:**
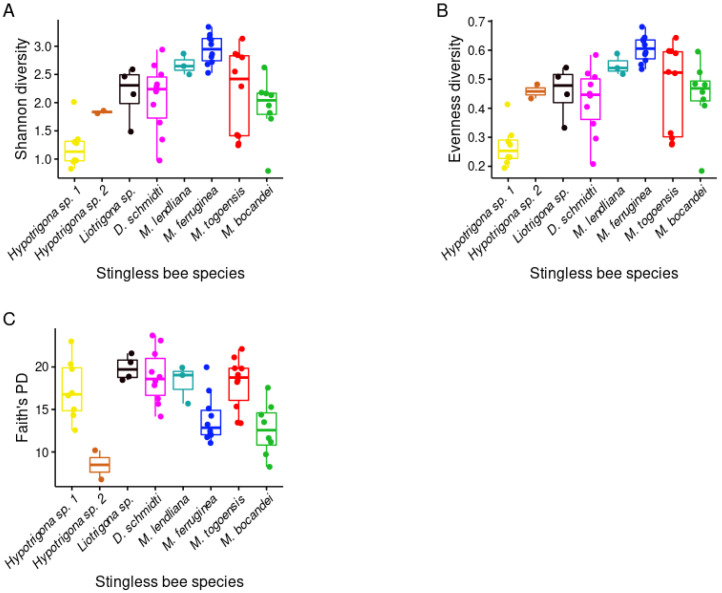
Alpha diversity estimates from the different stingless bee species. (**A**) Shannon diversity (*p* = 0.000038, *H* = 29.672), (**B**) Evenness diversity (*p* = 0.00005, *H* = 29.12), (**C**) Faith’s phylogenetic diversity (Faith’s PD) (*p* =0.00018, *H* = 24.888).

**Figure 3 microorganisms-09-02420-f003:**
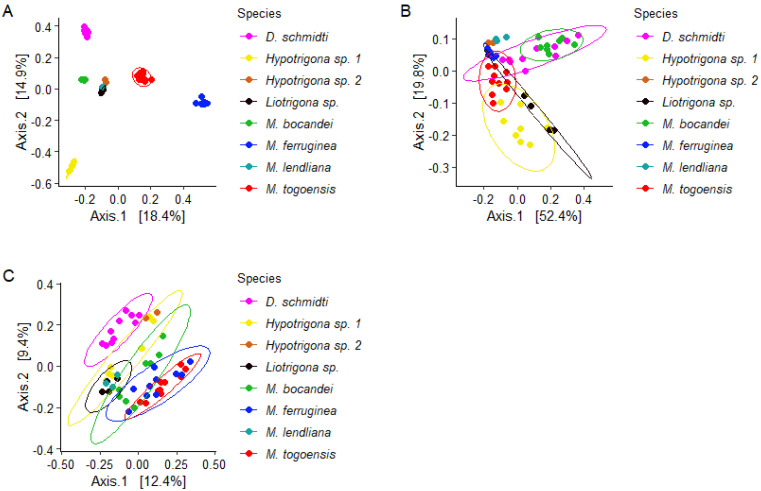
PCoA clustering using (**A**) Bray–Curtis PCoA, (**B**) Weighted UniFrac PCoA, (**C**) Unweighted UniFrac PCoA. The ellipse cluster indicate the distribution of the gut bacterial microbiota with respect to the different species of stingless bees (standard errors at 95% confidence).

**Figure 4 microorganisms-09-02420-f004:**
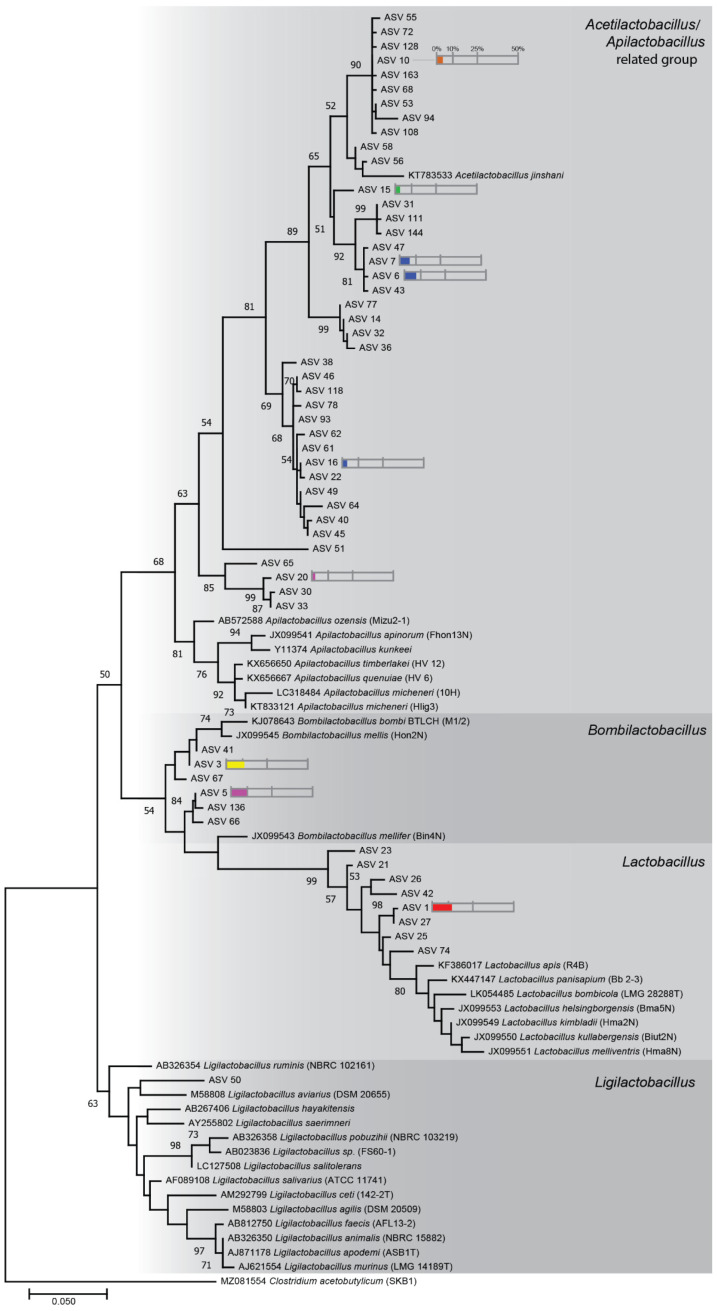
Phylogenetic tree showing the relationships between *Lactobacillaceae* ASVs from this study and reference *Lactobacillaceae* sequences. Accession Number are indicated at the beginning of each name reported from [41,44,55,56,57,58], and bacterial strains are indicated in brackets. Only ASVs that represent more than 1% among *Lactobacillaceae* family per species were represented. Percentage greater than 1% of the relative abundance from total reads of each ASV are represented by color bars. Each color represents each stingless bee species using the same color code as in Figure 2 and Figure 3.

**Figure 5 microorganisms-09-02420-f005:**
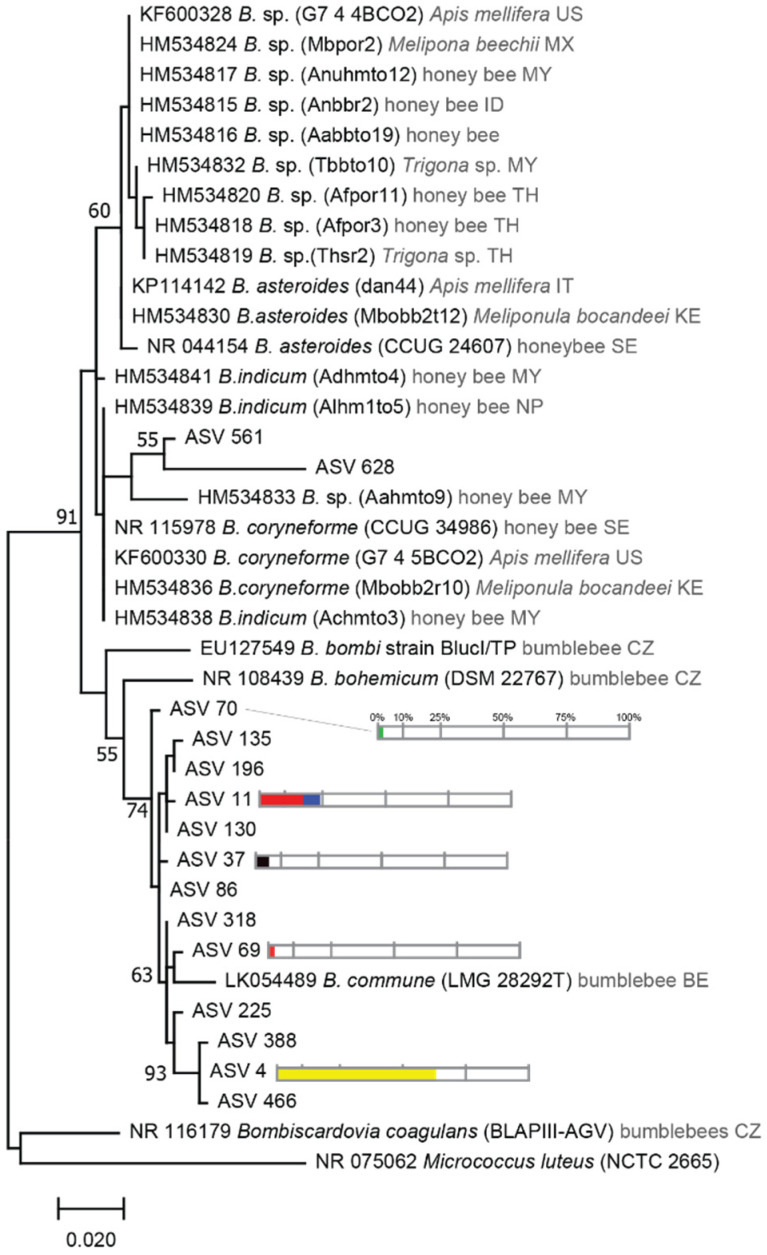
Phylogenetic tree showing the relationships between *Bifidobacterium* from stingless bees from Africa compared to other continents. Accession Number is indicated at the beginning of each name reported in [42,44,57,58,59,60,61,62]. Species from which the bacterial strain was reported and the country of origin (two letter code) are indicated in grey. Only ASVs that represent more than 1% among *Bifidobacteriaceae* family per species were represented. Percentage greater than 1% of the relative abundance from total reads of each ASV are represented by color bars. Each color represents each stingless bee species using the same color code as in Figure 2 and Figure 3.

**Figure 6 microorganisms-09-02420-f006:**
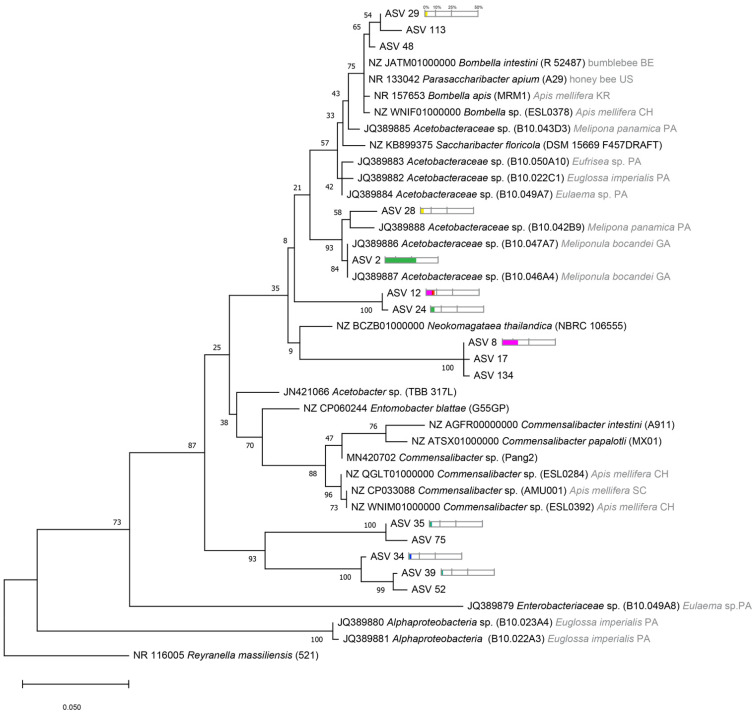
Phylogenetic tree showing the relationships between *Acetobacteraceae* family from African stingless bees compared to other continents. Accession Number is indicated at the beginning of each name reported in [41,42,61,63,64,66,67,68]. Species from which the bacterial strain was reported and the country of origin (two letter code) are indicated in grey. Only ASVs that represent more than 1% among *Acetobacteraceae* family per species are represented. Percentage greater than 1% of the relative abundance from total reads of each ASV are represented by color bars. Each color represents each stingless bee species using the same color code as in Figure 2 and Figure 3.

## Data Availability

The datasets presented in this study can be found in the following online repository PRJNA776526 and in Appendix A.

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
