# Peer review of "Loss and Gain of Gut Bacterial Phylotype Symbionts in Afrotropical Stingless Bee Species (Apidae: Meliponinae)"

_microorganisms, 2021, doi:10.3390/microorganisms9122420_

Round 1
Reviewer 1 Report
Advice: revision
The manuscript titled “Loss and gain of gut microbiota phylotype symbionts in Afrotropical stingless bee species (Apidae: Meliponinae) “ aims to understand host selective mechanisms that maintain distinct gut
communities among sympatric bee species. It can serve as a resource for future studies on bee health management and host-microbe co-evolution and adaptation. The overall manuscript was written and presented
well. The content is straightforward. Few shortcomings need to be resolved before considering publication.
Comments:
1. Since the article focuses on only bacterial diversity, it is better to mention gut bacteriome instead of gut microbes to justify the research work better.2. In my view, the introduction is very narrow and fails to contextualize the importance of insect symbiosis. It will be optimal if the authors consider adding one paragraph in the introduction about the importance of symbiosis in insects in general. Some key references are missing and can be added to enrich the introduction section:
a. Salem, H.; Kaltenpoth, M.: Beetle-bacterial symbioses: Endless forms most functional. Annual
Review of Entomology 67, pp. 201 - 219 (2022)
b. Ma, Qiuyu, et al. "Gut Bacterial Communities of Lymantria xylina and Their Associations with
Host Development and Diet." Microorganisms 9.9 (2021): 1860.
c. Chakraborty, A., & Roy, A. (2021). Microbial influence on plant–insect interaction. In Plant-Pest
Interactions: From Molecular Mechanisms to Chemical Ecology (pp. 337-363). Springer,
Singapore.
d. Schmidt, Konstantin, and Philipp Engel. "Mechanisms underlying gut microbiota–host interactions
in insects." Journal of Experimental Biology 224.2 (2021): jeb207696.
e. Chakraborty, Amrita, et al. "Unravelling the gut bacteriome of Ips (Coleoptera: Curculionidae:
Scolytinae): Identifying core bacterial assemblage and their ecological relevance." Scientific
Reports 10.1 (2020): 1-17.
f. Weisskopf, Laure, Stefan Schulz, and Paolina Garbeva. "Microbial volatile organic compounds in
intra-kingdom and inter-kingdom interactions." Nature Reviews Microbiology (2021): 1-14.
g. Näsvall, Karin, et al. "Host plant diet affects growth and induces altered gene expression and
microbiome composition in the wood white (Leptidea sinapis) butterfly." Molecular Ecology 30.2(2021): 499-516.
h. Duplais, Christophe, et al. "Gut bacteria are essential for normal cuticle development in herbivorous turtle ants." Nature communications 12.1 (2021): 1-6.
3. Line 104: is that OTU or ASV?
4. In the methodology section, it has been mentioned that bees were collected from different locations in
different years starting from 2015, and the samples were taken for gut dissection from the meliponary
from October 2019 to February 2020. I am curious to know how far prolonged exposure to a similar
controlled environment influences the gut microbiome? How far will the wild bee gut bacteriome be
different? It is interesting to have wild bee gut bacteriome inside the present study. Authors should
indicate the biases/ limitations clearly in the text.
5. Did you clean the bee gut contents before the DNA extraction? It was not mentioned in the text. The
DNA extraction section has minimal information. The authors should give some more details here,
like the DNA quality and quantity measurement method, before sending for sequencing.
6. How many biological replicates were used per bee sample for 16S? I did not find any statistics like
metastat /t-test regarding the differentially abundant bacteriome in different bees.
7. Please mention what the numbers mean in figure 1 B and C in the figure legend.
8. The authors should announce to the readers that the work reflects the diversity existing in the digestive
tract of the species, which does not mean that the taxa are metabolically active. The use of
metagenomic DNA instead of metagenomic RNA results in in-depth knowledge of the existing
microbiota but cannot recognize the metabolically active fraction. With this strategy, the
autochthonous microbiota of the digestive tract of bees cannot be distinguished from that associated
with the varied plant feeding/ foraging. Ideally, a metagenomic study of the bacteriome of insect food
(host plants) should have been carried out. The authors must recognize these limitations of the work,
and the readers are warned of these biases.
9. It will be interesting to predict the bacterial function in the bee gut by PICRUSt2 software.
# Douglas, G.M., Maffei, V.J., Zaneveld, J.R. et al. PICRUSt2 for prediction of metagenome functions.
Nat Biotechnol 38, 685–688 (2020). https://doi.org/10.1038/s41587-020-0548-6
10. PCoA clustering cannot rely much on fig 3 A and 3 C as it is based on one third or less microbiota
present in the samples.
11. The deposited data is not accessible to the reviewer. Please provide the link to evaluate.
12. Line 98: please correct the title. it should be “2.3. 16S rRNA”

Author Response
The manuscript titled “Loss and gain of gut microbiota phylotype symbionts in Afrotropical stingless bee species (Apidae: Meliponinae) “aims to understand host selective mechanisms that maintain distinct gut
communities among sympatric bee species. It can serve as a resource for future studies on bee health management and host-microbe co-evolution and adaptation. The overall manuscript was written and presented
well. The content is straightforward. Few shortcomings need to be resolved before considering publication.
We thank the Reviewer for all her/his constructive comments on our manuscript. Please find below point by point answer to each of her/his remarks.
Comments:
1. Since the article focuses on only bacterial diversity, it is better to mention gut bacteriome instead of gut microbes to justify the research work better.
R: We did not used “microbiome” in our manuscript, we believe the Reviewer refers to “microbiota”. We have changed “gut microbiota” for “gut bacterial microbiota” in the whole manuscript. We believe microbiome refers more to the coding capacity of the bacterial community which we do not address here in this study.
- In my view, the introduction is very narrow and fails to contextualize the importance of insect symbiosis. It will be optimal if the authors consider adding one paragraph in the introduction about the importance of symbiosis in insects in general. Some key references are missing and can be added to enrich the introduction section:
- Salem, H.; Kaltenpoth, M.: Beetle-bacterial symbioses: Endless forms most functional. Annual Review of Entomology 67, pp. 201 - 219 (2022)
- Ma, Qiuyu, et al. "Gut Bacterial Communities of Lymantria xylina and Their Associations with Host Development and Diet." Microorganisms 9.9 (2021): 1860.
- Chakraborty, A., & Roy, A. (2021). Microbial influence on plant–insect interaction. In Plant-Pest Interactions: From Molecular Mechanisms to Chemical Ecology (pp. 337-363). Springer, Singapore.
- Schmidt, Konstantin, and Philipp Engel. "Mechanisms underlying gut microbiota–host interactions in insects." Journal of Experimental Biology 224.2 (2021): jeb207696.
- Chakraborty, Amrita, et al. "Unravelling the gut bacteriome of Ips (Coleoptera: Curculionidae: Scolytinae): Identifying core bacterial assemblage and their ecological relevance." Scientific Reports 10.1 (2020): 1-17. f. Weisskopf, Laure, Stefan Schulz, and Paolina Garbeva. "Microbial volatile organic compounds in intra-kingdom and inter-kingdom interactions." Nature Reviews Microbiology (2021): 1-14.
- Näsvall, Karin, et al. "Host plant diet affects growth and induces altered gene expression and microbiome composition in the wood white (Leptidea sinapis) butterfly." Molecular Ecology 30.2(2021): 499-516.
- Duplais, Christophe, et al. "Gut bacteria are essential for normal cuticle development in herbivorous turtle ants." Nature communications 12.1 (2021): 1-6.
R: We have revised our introduction and added a general paragraph contextualizing the importance of insect gut microbiota in general. Additionally, we have included most of the references suggested by the Reviewer (L. 48-54).
- Line 104: is that OTU or ASV?
R: we thank the Reviewer for spotting this mistake, we have now made the change to ASV.
- In the methodology section, it has been mentioned that bees were collected from different locations in different years starting from 2015, and the samples were taken for gut dissection from the meliponary from October 2019 to February 2020. I am curious to know how far prolonged exposure to a similar controlled environment influences the gut microbiome? How far will the wild bee gut bacteriome be different? It is interesting to have wild bee gut bacteriome inside the present study. Authors should indicate the biases/ limitations clearly in the text.
We thank the Reviewer for this suggestion, it will be indeed very interesting to compare “semi-managed” stingless bees from the institute meliponary to feral colonies. Future studies will go in this direction. We have added few sentences in the discussion commenting on these limitations (L. 256-259).
- Did you clean the bee gut contents before the DNA extraction? It was not mentioned in the text. The DNA extraction section has minimal information. The authors should give some more details here, like the DNA quality and quantity measurement method, before sending for sequencing.
R: No, we used entire guts with their content in our experiments. We did not include too many details here since we are referring to our manuscript ([42] Tola, Y.H.; Waweru, J.W.; Hurst, G.D.D.; Slippers, B.; Paredes, J.C. Characterization of the Kenyan Honey Bee (Apis mellifera) Gut Microbiota: A First Look at Tropical and Sub-Saharan African Bee Associated Microbiomes. Microorganisms 2020, 8, 1721, doi:doi:10.3390/microorganisms8111721) where we describe everything in details. We have added, nonetheless, more information in M&M about the experimental procedures as suggested by the Reviewer (L. 93-100).
- How many biological replicates were used per bee sample for 16S? I did not find any statistics like metastat /t-test regarding the differentially abundant bacteriome in different bees.
The reviewer is requesting to analyze the abundance of the bacteriome among single bee samples; unfortunately, we cannot do such analysis because we used pooled samples in our analysis. Since most of the stingless bees are very small in size, and we were trying to characterize intra-species diversity rather than intra-bee diversity we opt for pooled samples. Pooling samples allowed us to reduce single bee diversity and cover more individuals in the whole analysis.
- Please mention what the numbers mean in figure 1 B and C in the figure legend.
R: We have now indicated what is the meaning of these numbers in Figure 1 legend. They correspond to the ASV ID number (L. 161-162).
- The authors should announce to the readers that the work reflects the diversity existing in the digestive tract of the species, which does not mean that the taxa are metabolically active. The use of metagenomic DNA instead of metagenomic RNA results in in-depth knowledge of the existing microbiota but cannot recognize the metabolically active fraction. With this strategy, the autochthonous microbiota of the digestive tract of bees cannot be distinguished from that associated with the varied plant feeding/ foraging. Ideally, a metagenomic study of the bacteriome of insect food (host plants) should have been carried out. The authors must recognize these limitations of the work, and the readers are warned of these biases.
R: We have now included these biases and limitations of our study in the Discussion part (L. 273-280).
- It will be interesting to predict the bacterial function in the bee gut by PICRUSt2 software. # Douglas, G.M., Maffei, V.J., Zaneveld, J.R. et al. PICRUSt2 for prediction of metagenome functions. Nat Biotechnol 38, 685–688 (2020). https://doi.org/10.1038/s41587-020-0548-6
R: We fully agree with the Reviewer that performing bacterial function prediction using PICRUSt2 would be an important addition to our paper. Unfortunately, we did not have time during the 10 days review time limit. Additionally, the v3-v4 16S rRNA data has not the enough resolution for a functional analysis; as a result, we are planning to do transcriptomic analysis to validate 16S data for the functional prediction as the next experiment.
- PCoA clustering cannot rely much on fig 3 A and 3 C as it is based on one third or less microbiota
present in the samples.
R/ The Reviewer may have misunderstood or mentioned microbiota instead of stingless bee samples. We included all our data and all the samples we analyzed in this study.
- The deposited data is not accessible to the reviewer. Please provide the link to evaluate.
R: We apologize, we have now made the data accessible (PRJNA776526).
- Line 98: please correct the title. it should be “2.3. 16S rRNA”
R: Corrected.
Reviewer 2 Report
The authors ask about the gut microbiota of eight selected Afrotropical stingless bee species. As mentioned in the submitted manuscript, stingless bees could be recognized as economically important species, as they could be used as crop pollinators, as well as honey, propolis, and wax producers.
In the submitted manuscript, the authors determined profiles of bacterial communities associated with selected species using the V3-V4 fragment of the 16S gene as a marker. I agree that the research questions are current and valuable. Moreover, the insight into the microbiota of selected species may bring important data for further, more comprehensive studies for this group of insects.
I have only a few remarks that I believe will help to improve the overall presentation of the submitted manuscript. First of all, I am wondering why the authors assumed that Firmicutes, Actinobacteria, and Proteobacteria are conserved phyla across selected species. In fact, those phyla are commonly found as the most abundant bacteria in bacterial communities of insects in general. Moreover, could you add the fragment describing the quality and quantity check of extracted DNA? I am wondering if e.g. NanoDrop or Qubit was used or any other method. My next comment concerns the information about the selected region of 16S. The details are described in Results, but in my opinion, they should be also added to the Methods. The last question concerns the heatmaps shown in the Results section. The colors correspond to the average relative abundances, but the scale is missed. I am wondering whether those colors correspond to the same values of abundance.
I have also a few minor remarks:
- line 41 – italics is missing (Meliponula);
- line 46 – an additional comma is added (e.g.,);
- line 49 – “an” instead of “and”?;
- line 87 – a missing genus of the crop? (two commas in parentheses);
- line 98 – 16S is misspelled in the section header;
- line 140 – missing comma (, and Acetobacter);
- line 224 – missing first parenthesis;
- line 242 – an additional space before the comma;
- lines 246-247 – italics is missing (names of bacterial genera);
- line 261 – “correlation” instead of “correlated”?
To sum up, I recommend the submitted manuscript for publication after minor revision.
Author Response
Comments and Suggestions for Authors
The authors ask about the gut microbiota of eight selected Afrotropical stingless bee species. As mentioned in the submitted manuscript, stingless bees could be recognized as economically important species, as they could be used as crop pollinators, as well as honey, propolis, and wax producers.
In the submitted manuscript, the authors determined profiles of bacterial communities associated with selected species using the V3-V4 fragment of the 16S gene as a marker. I agree that the research questions are current and valuable. Moreover, the insight into the microbiota of selected species may bring important data for further, more comprehensive studies for this group of insects.
I have only a few remarks that I believe will help to improve the overall presentation of the submitted manuscript. First of all, I am wondering why the authors assumed that Firmicutes, Actinobacteria, and Proteobacteria are conserved phyla across selected species. In fact, those phyla are commonly found as the most abundant bacteria in bacterial communities of insects in general. Moreover, could you add the fragment describing the quality and quantity check of extracted DNA? I am wondering if e.g. NanoDrop or Qubit was used or any other method. My next comment concerns the information about the selected region of 16S. The details are described in Results, but in my opinion, they should be also added to the Methods. The last question concerns the heatmaps shown in the Results section. The colors correspond to the average relative abundances, but the scale is missed. I am wondering whether those colors correspond to the same values of abundance.
We thank the reviewer for his valuable comments about our manuscript.
We agreed with the Reviewer that Firmicutes, Actinobacteria, and Proteobacteria are highly associated with insects. Nonetheless, not every insect has a resident gut microbiota, nor the three phyla, always, in their gut. Multiple studies have shown that these three groups are always present in corbiculate bees, and more specifically in stingless bees. We revised again the whole manuscript and we believe that we have not stated that these are the only group of insects that are associated with these bacteria. Unfortunately, the resolution of the 16S V3-V4 region and the actual taxonomy of these phyla found in the literature does not allow us to be more precise and maybe be able to name the conserved genera.
We have added in M&M the information about DNA concentration and quality check. This information was previously found in the reference from our paper that we have in this session. ([42] Tola, Y.H.; Waweru, J.W.; Hurst, G.D.D.; Slippers, B.; Paredes, J.C. Characterization of the Kenyan Honey Bee (Apis mellifera) Gut Microbiota: A First Look at Tropical and Sub-Saharan African Bee Associated Microbiomes. Microorganisms 2020, 8, 1721, doi:doi:10.3390/microorganisms8111721). We used NanoDrop (L 98-100). Of note, when Macrogen receive the samples in their facilities, they conducted a further concentration and quality analysis using DNA QC-Picogreen.
As suggested by the Reviewer we moved the information about the 16S region sequenced to M&M.
We thank reviewer for spotting that the color scale in Figure 1 was missing. The numbers above the heat map correspond to the ASV ID number and not to their abundances. We have now included the scale, information in Figure 1, and the text in Figure and Figure legend to clarify this misunderstanding (L. 161-162).
I have also a few minor remarks:
- line 41 – italics is missing (Meliponula);
- line 46 – an additional comma is added (e.g.,);
- line 49 – “an” instead of “and”?;
- line 87 – a missing genus of the crop? (two commas in parentheses);
- line 98 – 16S is misspelled in the section header;
- line 140 – missing comma (, and Acetobacter);
- line 224 – missing first parenthesis;
- line 242 – an additional space before the comma;
- lines 246-247 – italics is missing (names of bacterial genera);
- line 261 – “correlation” instead of “correlated”?
To sum up, I recommend the submitted manuscript for publication after minor revision.
We have corrected all minor remarks raised by the Reviewer.
Round 2
Reviewer 1 Report
The authors answer most of the issues nicely.
I still believe they need to conduct a metastat or similar analysis between different bee species to find differentially abundant bacteria (if any) among eight Afrotropical stingless bee species. This will improve the quality of the manuscript substantially.